# Error Estimation of Ultra-Short Heart Rate Variability Parameters: Effect of Missing Data Caused by Motion Artifacts

**DOI:** 10.3390/s20247122

**Published:** 2020-12-11

**Authors:** Alessio Rossi, Dino Pedreschi, David A. Clifton, Davide Morelli

**Affiliations:** 1Department of Computer Science, University of Pisa, 56126 Pisa, Italy; dino.pedreschi@unipi.it; 2Department of Engineering Science, Institute of Biomedical Engineering, University of Oxford, Oxford OX1 2JD, UK; david.clifton@eng.ox.ac.uk (D.A.C.); davide.morelli@eng.ox.ac.uk (D.M.); 3Huma Therapeutics Limited, London SW1P 4QP, UK

**Keywords:** SDNN, rMSSD, HRV, autonomic nervous system

## Abstract

Application of ultra–short Heart Rate Variability (HRV) is desirable in order to increase the applicability of HRV features to wrist-worn wearable devices equipped with heart rate sensors that are nowadays becoming more and more popular in people’s daily life. This study is focused in particular on the the two most used HRV parameters, i.e., the standard deviation of inter-beat intervals (SDNN) and the root Mean Squared error of successive inter-beat intervals differences (rMSSD). The huge problem of extracting these HRV parameters from wrist-worn devices is that their data are affected by the motion artifacts. For this reason, estimating the error caused by this huge quantity of missing values is fundamental to obtain reliable HRV parameters from these devices. To this aim, we simulate missing values induced by motion artifacts (from 0 to 70%) in an ultra-short time window (i.e., from 4 min to 30 s) by the random walk Gilbert burst model in 22 young healthy subjects. In addition, 30 s and 2 min ultra-short time windows are required to estimate rMSSD and SDNN, respectively. Moreover, due to the fact that ultra-short time window does not permit assessing very low frequencies, and the SDNN is highly affected by these frequencies, the bias for estimating SDNN continues to increase as the time window length decreases. On the contrary, a small error is detected in rMSSD up to 30 s due to the fact that it is highly affected by high frequencies which are possible to be evaluated even if the time window length decreases. Finally, the missing values have a small effect on rMSSD and SDNN estimation. As a matter of fact, the HRV parameter errors increase slightly as the percentage of missing values increase.

## 1. Introduction

Heart Rate Variability (HRV) is a physiological phenomenon of fluctuation between adjacent Inter-Beat Intervals (IBI) [1,2]. The heart beats (HR) average and HRV are regulated by two main mechanisms. First, they are controlled by Autonomic Nervous Systems (ANS), which affect the activity of the sinoatrial node (SAN) [3,4,5]. In particular, ANS controls SAN by two signaling pathways that permit increasing a decrease in HR by sympathetic and parasympathetic nervous systems stimulations, respectively [2]. The second mechanism that regulates HR and HRV is the *coupled-clock* system which drives the automaticity of human SAN pacemaker cells even without neural input [3,6]. Aging and clinical condition (e.g., cardiovascular disease) affect both ANS and SAN responses inducing an alteration in HR and HRV [2,3]. For this reason, monitoring HR and HRV response permits assessing possible deterioration of cardiovascular responses. In particular, abnormal hourly heart rate patterns associated with an increase in HRV values have been found to be strongly correlated to an increased risk of mortality (particularly among the elderly) causing, for example, atrial fibrillation [7]. Moreover, HRV parameters are useful parameters to provide insight about sympathetic and parasympathetic nervous system responses of cardiac vagal tone that was found to be linked to cognitive, emotional, social, and health status [8].

SDNN, i.e., standard deviation of the normal-to-normal interval between QRS complexes, rMSSD, i.e., the root mean squared of successive inter-beat intervals, are the two widely used HRV parameters to predict many health issues [2]. SDNN is considered the “gold standard” for cardiovascular risks when recorded over 24 h (SDNN24) [9]. As a matter of fact, it has been found that SDNN24 values lower than 50 milliseconds (ms), between 50 and 100 ms and higher than 100 ms, indicate people with unhealthy, compromised health and healthy status, respectively [10]. In particular, Keiger et al. [10] demonstrated that people with SDNN24 higher than 100 ms have a 5.3 times lower risk of mortality at follow-up compared to those with SDNN24 lower than 50 ms. Moreover, it was found that SDNN24 values lower than 80 ms are predictive of cardiac events [11,12]. On the contrary, RMSSD reflects the IBI variance that is sensitive to high-frequency (HF) heart period fluctuations such as in the respiratory frequency range and is used as an index of vagal cardiac control [13]. It is considered the most useful marker for the parasympathetic neural system activation [9,14]. RMSSD is widely used by professional athletes in order to monitor cardiac activity and consequently their health status. In particular, it has been found that acute decreases in HRV have been linked to intense endurance training [15], resistance training [16], and competition [17]. Therefore, low HRV reflects acute fatigue from high intensity physical effort. Moreover, reduced resting HF is also linked to altered activation of prefrontal cortex and consequently to low cognitive functions that were found to be associated with stress, panic, anxiety, or worry [18]. Actually, in the last few decades, researchers investigated the association between HRV and depression reporting a strong association between reduced rMSSD and depression [19,20].

HRV has been traditionally obtained from 5 min or 24 h IBI time series [2]. Thanks to the recent technological advancements of wrist-worn wearable devices equipped with a heart rate sensor, it is now possible to semi-continuously obtain passive cardiac response measurements during people’s daily lives. For this reason and because of the low cost of these devices, they might have a great impact on the medical field due to the fact that it is possible to assess HRV features 24 h per day, 7 days per week [21]. The main problem of wrist-worn wearable devices is the artifacts induced by external stimuli that produce inconsistent IBI values. Usually, when IBI are recorded with gold standard technology (i.e., electrocardiography), the number of abnormal values is close to 1% [22], while, when it is recorded by these low cost instruments, is more than 10% [23]. Hence, estimating the error caused by this huge quantity of missing values is fundamental to obtain reliable values from wrist-worn wearable devices equipped with heart rate sensors avoiding misleading results [24]. To obtain reliable HRV data from these devices, shorter recordings (lower than 5 min) of IBI data during real-life scenarios are required. The ultra-short HRV parameters permit reducing the influence of motion artifacts on data recording due to the low time window length and consequently low percentage of missing values. Several studies estimate the reliability of HRV analysis from IBI time series shorter than 5 min for the clinical purposes demonstrating that 10 and 30 s recordings are the minimum time required to obtain accurate measures of RMSSD and SDNN, respectively [25,26,27,28]. However, to the best of our knowledge, no study are conducted in order to estimate the effect of missing values induced by wrist-worn devices equipped with HR sensors on ultra-short HRV parameters.

Hence, the aim of this study is to quantify the error induced by ultra-short IBI time series (i.e., from 4 min to 30 s) and missing values (from 0 to 70%) in order to detect if it is possible to accurately estimate SDNN and rMSSD by using wrist-worn wearable devices. To this aim, we simulate missing values induced by motion artifacts by a Gilbert burst model which creates a burst-error with a two-state Markov chain in 22 young healthy subjects.

## 2. Materials and Methods

### 2.1. Data

In this paper, we used the Multilevel Monitoring of Activity and Sleep in Healthy people (MMASH) dataset [29,30,31] providing 24 h of continuous Inter-Beat Intervals data (*IBI*), triaxial accelerometer data, sleep quality, physical activity, and psychological characteristics (i.e., anxiety status, stress events, and emotions) for 22 healthy young males (age = 27.29 ± 4.21 yrs; height = 179.91 ± 8.22 cm; weight = 75.05 ± 12.79 kg). Participants’ anthropomorphic characteristics (i.e., age, height, and weight) were recorded at the start of the data recording. Moreover, participants filled in questionnaires that provide information about participants’ psychological status, i.e., chronotype, anxiety status, and sleep quality. During the 24 h of the data recording, participants wore two devices that continuously recorded heart response (Polar H7 heart rate monitor—Polar Electro Inc., Bethpage, NY, USA) and the actigraphy data (ActiGraph wGT3X-BT—ActiGraph LLC, Pensacola, FL, USA). Moreover, the perceived moods were recorded at different times of the day and the daily stress was provided before sleeping in order to summarize the individual’s stressful events of the day. Finally, twice a day (i.e., before going to bed and when they woke up), the subjects collected saliva samples at home in appropriate vials in order to assess Melatonin and Corisol saliva concentration. More details about the experimental setup of the MMASH dataset are provided in the data descriptor paper written by Rossi et al. and published on MDPI Data journal in 2020 [29].

In this study, we used only *IBI* data continuously recorded over 24 h on 22 healthy young males by a Polar H7 chest strap (Polar Electro Inc., Bethpage, NY, USA). This device is a Bluetooth low energy chest strap with an ECG sensor that provides valid information of inter-beat intervals (*IBI*) [32]. Participants wore the Polar H7 chest strap for 24 h (between 9:00 a.m. and 9:00 p.m. on the next day), and were instructed to wear the chest straps during both day (during physical activities too) and night. For the aim of this study, the IBI timeseries of each participants were split into 5 min time windows in order to compute gold standard HRV parameters. Moreover, to assess the effect of length of the time window on HRV estimation, ultra-short time windows (from 4 min to 30 s) were selected in each 5 min time window, and the HRV parameters were computed. Finally, we introduce missing values to assess their effect on HRV features. More details about the dataset preprocessing and the HRV estimation were provided in Section 2.2 and Section 2.3, respectively. Finally, the data analysis details are provided in Section 2.4.

In accordance with the Helsinki Declaration as revised in 2013, the study was approved by the Ethical Committee of the University of Pisa (#0077455/2018).

### 2.2. Data Preprocessing

First of all, by using the Python *hrv-analysis* library (https://pypi.org/project/hrv-analysis), 2.18 ± 1.29% of RR-intervals in the dataset were detected as ectopic beats (i.e., disturbance of the cardiac rhythm frequently related to the electrical conduction system of the heart) or missing values induced by motion artifacts. These missing values were reconstructed via quadratic interpolation applied on the time domain, i.e., the heartbeats timestamp, instead of the duration domain, i.e., the duration of the heartbeats, as suggested by Morelli et al. [23].

In order to compute gold standard SDNN and rMSSD HRV features, the participants’ 24 h RR-intervals time series were split into 5 min windows (6130 5 min time windows were extracted). Moreover, to detect the influence of ultra-short HRV features in each 5 min window, ultra-short windows (4 min, 3 min, 2 min, 1 min and 30 s) were randomly selected.

Finally, in order to simulate motion artifacts observable during *IBI* recording by wrist-worn devices equipped with PPG sensors, artificial missing values (i.e., 5%, 10%, 15%, 30%, 50%, and 70%) were created for each 5 min and ultra-short time windows in accordance with a random walk Gilbert burst model which simulates burst-error with a two-state Markov chain (i.e., good as 0 and bed as 1) [33] as showed in Morelli at al.’s paper [23].

### 2.3. HRV Parameters

HRV is usually calculated from 5 min *IBI* time series [34]. For the aim of our study, we focused on the two most popular time-domain HRV measurements:SDNN refers to the standard deviation of *IBI*. It estimates overall power spectrum of *IBI* timeseries. The SDNN is defined as the “gold standard” to assess both morbidity and mortality in the population [2].rMSSD is the root mean square of the successive *IBI* differences estimates short-term components of HRV [35].

These two HRV metrics are computed in each 5 min window and in each ultra-short window with and without missing values.

Moreover, to better investigate the effect of missing values in the ultra-short HRV parameters, the power spectrum of the *IBI* time series was computed. The power spectrum $S_{xx}\left(f\right)$ of a time series x(t) describes the distribution of power into frequency components (ω) composing that signal. The minimal frequency that is possible to observe for each ultra-short time series was defined as a ratio between 1 and its length expressed in seconds (1/s), while the maximum is set at 0.4 in accordance with the literature. In order to derive the Power Spectrum Density (PSD) of the time series x(t), the Lomb–Scargle Periodogram was used instead of Fourier transformation because the inter-beats intervals (IBI) are not uniformly distributed [36,37]. VLF (power in very low-frequency ranges, i.e., ≤0.04 Hz), LF (power in low-frequency ranges, i.e., 0.04–0.15 Hz), HF (Power in high-frequency ranges, i.e., 0.15–0.4 Hz), and total power (Power in all the frequency ranges, i.e., ≤0.4) were obtained by the integration of the power in the relevant frequency range in the spectrum as showed in Equation (Equation 1). Finally, the ratio between LF and HF (LF/HF) was also computed:(1)$$PSD_{freq}=\int_{min_{freq}}^{max_{freq}}S_{xx}\left(\omega \right)d\omega$$

### 2.4. Data Analysis

The relationship between “gold standard” HRV parameters computed in a 5 min window and ultra-short HRV parameters with and without missing values are assessed by Pearson’s correlation coefficient. Values higher than 0.8 reflect a strong correlation. The estimation error between HRV parameters compute on 5 min and in ultra-short time window with and without missing values were computed by the Root Mean Squared Error (RMSE). The magnitude of the difference was assessed by Cohen’s d effect size (ES). It is computed as the ratio between the mean difference between HRV gold standard parameters and ultra-short ones, and the pooled standard deviation as showed in Equation (Equation 2). ES < 0.2, (0.2–0.5], (0.5–0.8], >0.8 are considered trivial, small, medium, and large effect size. For example, if the ES is equal to 0.04, the two groups’ means differ by 0.04 standard deviations, and this difference should be considered trivial:(2)$$ES=\frac{\bar{HRV_{5min}}-\bar{HRV_{ultra-short}}}{(\sigma {}_{HRV_{5min}}+\sigma {}_{HRV_{ultra-short}})/2}$$

Bland and Altman plot (BA)was assessed in order to quantify the agreement between two quantitatives. Due to the fact that a huge number of Bland–Altman plots are required to describe all the comparisons, we decided to not insert all of these plots. We have resumed the results of BA plots by Bias, i.e., mean ± standard deviation of the differences, and systematic error, i.e., Pearson’s correlation coefficient between difference and means, of ultra-short HRV parameters with and without missing values compared to “gold standard” HRV parameters.

Finally, Spearman’s rank correlation coefficient was computed in order to assess the relationship between HRV features.

All the data preprocessing, feature extraction and data analyses are conducted by using the Python 3 programming language.

## 3. Results

Descriptive statistics of all the participants’ psycho-physiological features provided in MMASH dataset are provided in Rossi et al.’s data descriptor paper [29].

Table 1 and Table 2 report the descriptive statistics and statistical results of SDNN and rMSSD for all of the windows and missing values percentage, respectively. Small ES and moderate correlation (i.e., ES < 0.38, r > 0.69; see Table 1) are detected for ultra-short SDNN values with a time window longer than 2 min showing a bias lower than 10.70 ms (RMSE < 26.84 ms) without a statistically significant systematic error. On the contrary, moderate ES (i.e., ES > 0.45) is detected for time windows shorter than 1 min. On the contrary, rMSSD shows a trivial ES and moderate/strong correlation (i.e., ES < 0.20, r > 0.75; see Table 2) for almost all of the time windows showing a bias lower than 4.83 ms (RMSE < 21.47 ms) with a small systematic error (not statistically significant). In particular, only 70% of the missing values in 1 min and 30 s windows show small ES and moderate correlation (i.e., ES > 0.20, r < 0.62) with a bias higher than 5.37 ms (RMSE > 24.01 ms), while other time series lengths with a different percentage of missing values show trivial ES.

Figure 1 compares the power spectrum density (PSD) of the 5 min time series and the ultra-short ones with missing values. The lower the time series length, the lower the very low frequencies (VLF) that are possible to evaluate. In particular, frequencies lower than 0.004 (1/240 s), 0.006 (1/180 s), 0.008 (1/120 s), 0.017 (1/60 s), and 0.03 (1/30 s) Hz are not possible to evaluate for 4 min, 3 min, 2 min, 1 min, and 30 s, respectively. However, even if the time series decrease, it is possible to assess both LF and HF. Due to the fact that rMSSD reflects the HF (a strong correlation was detected between rMSSD and HF in 5 min time series—Table 3), it is possible to accurately estimate rMSSD even if the length of the time series decreases due to the fact that all HF could be assessed. On the contrary, missing very low frequencies negatively affect SDNN estimation as the length of the time series decreases. In particular, the moderate correlation detected between SDNN and Total power (Table 3) demonstrates that short time series do not permit assessing all the frequencies in the power spectrum negatively affecting the SDNN estimation.

In line with the results found in Table 1 and Table 2 for SDNN and rMSSD, respectively, Table 4 shows trivial and small ES for LF and HF for all of the ultra-short time series with almost all of the missing values (except for 30 s time series with 70% of missing value). Moreover, trivial and small ES for time series longer than 1 min and of 4 min were detected for total power and VLF, respectively. In particular, only total power of 4 min time series with all of the missing values percentage and 3 min time series with 0% of missing values show a trivial ES. Additionally, Table 4 and Figure 1 show that VLF and LF are underestimated as the percentage of missing values increases, while HF is overestimated. For these reasons, the length of the time series and the missing values strongly affect LF/HF HRV parameters. As a matter of fact, it is possible to estimate LF/HF values only in 4 min time windows without missing values where LF and HF are accurately detected.

## 4. Discussion

HRV is a widely used tool in many research areas, e.g., cardiovascular, preventive medicine, and sport [38,39,40,41,42,43,44], due to its non-invasive and reliable characteristics that permit assessing ANS and SAN activation. Nowadays, long- and short-term recordings (i.e., 24 h and 5 min, respectively) are considered the more reasonable options for HRV analysis. However, in real-world applications, shorter IBI recordings than 5-min are required which permit the mobile devices to instantaneously display results avoiding any possible misleading insight induced by motion artifacts. Therefore, representative studies are required in literature to provide reference values for short-term HRV analysis with missing values. The main finding of this study is that our results are in line with the ones reported for other previous studies [25,26,27,28]. As a matter of fact, we found that rMSSD requires at least 30 s to obtain reliable values (trivial effect size; Table 2). On the contrary, SDNN requires IBI timeseries longer than 2 min to have reliable results (small effect size; Table 1). Moreover, the novel results provided in this study are that rMSSD is more highly affected by missing values compared to SDNN (i.e., 0.12 ± 0.04 and 0.04 ± 0.03, respectively) showing, however, a small increment in effect size from all of the time series lengths as their percentage of missing values increase.

To truly estimate SDNN, we need precise information from the whole power spectrum, i.e., from VLF to HF [45]. As a matter of fact, we have found a moderate/strong relationship (r = 0.73) between total power and SDNN obtained from 5 min time series. In particular, SDNN is more correlated with VLF than with LF and HF (Table 3), and the power of VLF (49.92%) is larger than the power of LF (31.88%) and HF (18.20%) by several orders of magnitude (Table 4) resulting in a smaller contribution of HF and LF to SDNN than VLF [45]. Hence, the small ES and RMSE higher than 22.88 ms between SDNN obtained from 5 min time series and ultra-short ones are induced by the fact that it is not possible to evaluate all the VLF required to estimate SDNN (Figure 1). For example, in 2 min IBI time series, it is possible to evaluate the power of frequencies higher than 0.008 losing the information of frequencies between 0.0033 and 0.008. For this reason, SDNN shows a bias of about −9.51 ± 23.28 ( RMSE = 25.15) with a small effect size (ES = 0.34). For these reasons, SAN is lower reflected in ultra-short SDNN estimation compared to 5-min time window ones. Moreover, a very small change in ES (0.04 ± 0.03) between 0 to 70% of missing values in all the time windows demonstrated that SDNN is accurate even if motion artifacts affect the IBI time series. Hence, SDNN is more affected by the length of the time window than the missing values that could be detected throughout the IBI time series.

On the contrary, rMSSD is more sensitive to high-frequency (HF) fluctuations [13,45] compared to LF, VLF, and total power. As a matter of fact, strong correlation is detected between rMSSD and HF obtained in 5 min IBI time series (r = 0.80, Table 3). Hence, even if the length of the IBI time series decreases, it still possible to accurately assess the HF. As a matter of fact, Beak et al. [26] have found that rMSSD and in particular HF calculated using ultra-short IBI time series could be a good surrogate to those calculated using standard 5-min intervals. For this reason and as shown in Table 2, rMSSD is found to be accurate for all the IBI timeseries without missing values (ES < 0.11, RMSE < 15.13 ms, r < 0.79 and bias <−2.34 ms). For all of the time windows, ES shows a small increment with the missing values up to 50% (0.05 ± 0.02), while an increase of about 0.15 ± 0.03 for rMSSD estimated with 70% of missing values. However, trivial ES was observed for almost all the time windows with 70% of missing values. As a matter of fact, Table 4 and Figure 1 show trivial ES in HF for up to 50% of missing values for all the time windows, while a small ES was detected for time windows with 70% of missing values shorter than 1 min. On the contrary, missing values and the length of the time windows strongly affect the ratio between HF and LF. In particular, only a 4 min time window without missing values permits accurately estimating LF/HF HRV parameters.

These results highlight the fact that approaches to interpolating missing values are not required for rMSSD, but they are needed for SDNN estimation as also suggested by Morelli et al. [23]. As a matter of fact, rMSSD captures fast changes in heart activity (HF), while SDNN is an index of the entire power spectrum and in particular of VLF (Table 3 and Table 4). For this reason, interpolation methods that act as low pass filters affecting the power spectrum density of IBI time series have a negative impact on rMSSD, while a small effect on SDNN [23]. On the contrary, no interpolation approach introduced white errors, thus minimizing the impact on IBI successive differences that were the first computation step of rMSSD. Moreover, due to the fact that SDNN can be estimated from the entire power spectrum [45], the estimation of missing frequencies is required to improve accuracy of SDNN. Future works are scheduled in order to assess the effect of interpolation approaches and missing frequencies’ estimation from known ones. Moreover, due to the fact that low cost wearable devices equipped with heart rate sensors are becoming more and more accurate in IBI recording, the reduction of noisy data could be the right choice to improve the quality of HRV parameters’ estimation. However, this paper suggests that, even if the IBI time series contain a large amount of noise, a small error is expected in the estimation of both SDNN and rMSSD. In conclusion, wrist-worn low-cost devices could be a good surrogate of a good quality simple ECG even if they could have a huge quantity of missing values.

A limitation of this study is that the participants are all young males. Future works could be performed in order to detect if individual characteristics such as gender, age, and individual’s habits could affect the estimation of ultra-short SDNN and rMSSD with and without missing values. Moreover, due to the fact that SDNN24 is an important HRV parameter to detect people’s health status, an investigation is required to assess the effect of missing values on this HRV parameter.

## 5. Conclusions

As showed in previous studies, it is possible to accurately estimate rMSSD and SDNN by using 30 s and 2 min time windows, respectively. In particular, in this paper, we highlight the fact that ultra-short IBI time series do not permit assessing all the VLF increasing the bias on SDNN. Moreover, to the best of our knowledge, this is the first study that evaluates the effect of missing values on ultra-short HRV data. We found that the missing values have a low effect on both SDNN estimations, while having a moderate effect on rMSSD. In particular, the higher the missing values, the higher the bias in both rMSSD and SDNN showing trivial and small magnitudes of the differences, respectively.

## Figures and Tables

**Figure 1 sensors-20-07122-f001:**
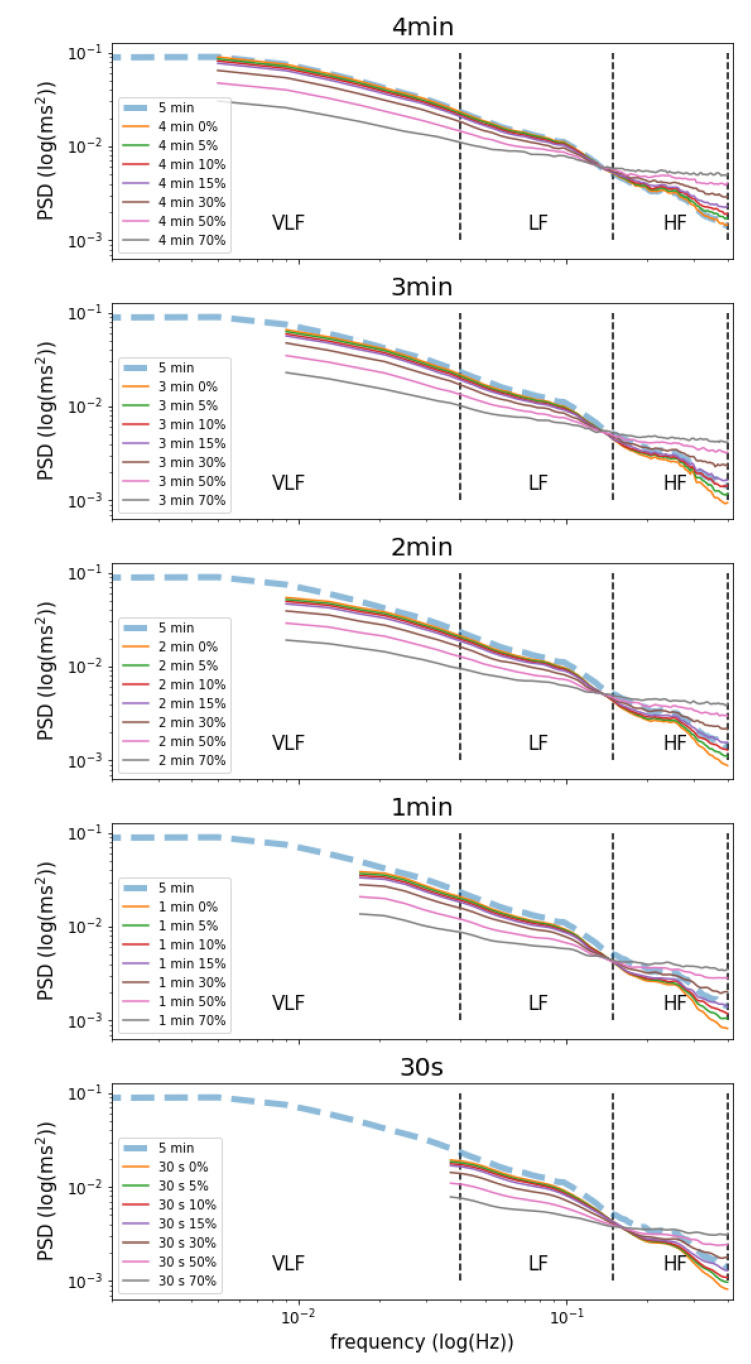
Power Spectrum Density plot for timeseries with different time series length and missing values percentage. The blue dotted line in each plot reflects the power spectrum of the 5 min time window.

**Table 1 sensors-20-07122-t001:** Descriptive statistics and statistical analysis of SDNN for all of the time window and missing values. The statistical analysis is focused to compare 5 min HRV parameters to ultra-short ones. * refers to small ES of ultra-short HRV features compared to 5 min ones, respectively. r, RMSE, and ES refer to the correlation, the Root Mean Square Error, and the effect size between the ‘gold standard’ and the ultra-short with and without missing values on SDNN, respectively. Bias refers to mean ± standard deviation of the difference of ultra-short HRV parameters compared to ‘gold standard’ HRV parameters (i.e., 5 min time window), while systematic error is the Pearson’s correlation coefficient between means and differences between 5 min and ultra-short HRV estimation.

Time Window	Missing Values	SDNN	r	RMSE	Bias	Systematic Error	ES
5 min	—	72.70 ± 30.23	—	—	—	—	—
4 min	0%	69.32 ± 35.83	0.79	23.17	−6.92 ± 22.11	0.08	0.20 *
	5%	69.15 ± 36.30	0.79	23.20	−6.94 ± 22.17	0.08	0.20 *
	10%	69.29 ± 36.42	0.79	23.25	−7.02 ± 22.31	0.08	0.20 *
	15%	69.18 ± 36.36	0.78	23.31	−7.08 ± 22.45	0.08	0.21 *
	30%	69.02 ± 36.16	0.78	23.87	−7.20 ± 22.75	0.08	0.21 *
	50%	68.62 ± 36.26	0.77	24.71	−7.48 ± 23.55	0.09	0.21 *
	70%	67.64 ± 36.66	0.74	26.02	−8.27 ± 24.67	0.10	0.23 *
3 min	0%	65.96 ± 31.99	0.76	22.88	−7.64 ± 21.57	0.04	0.26 *
	5%	65.92 ± 31.97	0.76	22.95	−7.66 ± 21.59	0.04	0.26 *
	10%	65.9 ± 31.79	0.76	22.98	−7.69 ± 21.61	0.04	0.26 *
	15%	65.79 ± 31.95	0.76	23.06	−7.73 ± 21.65	0.04	0.26 *
	30%	65.72 ± 32.18	0.76	23.19	−7.84 ± 21.82	0.05	0.26 *
	50%	65.31 ± 32.33	0.75	23.71	−8.19 ± 22.25	0.05	0.27 *
	70%	64.54 ± 32.98	0.73	24.65	−8.76 ± 23.04	0.08	0.29 *
2 min	0%	63.57 ± 31.39	0.71	25.15	−9.51 ± 23.28	0.03	0.34 *
	5%	63.52 ± 31.42	0.71	25.19	−9.56 ± 23.51	0.03	0.34 *
	10%	63.5 ± 31.37	0.71	25.25	−9.59 ± 23.57	0.03	0.35 *
	15%	63.4 ± 31.44	0.71	25.31	−9.60 ± 23.55	0.04	0.35 *
	30%	63.22 ± 31.56	0.70	25.66	−9.76 ± 23.73	0.04	0.35 *
	50%	62.82 ± 31.85	0.70	25.98	−10.60 ± 23.95	0.05	0.37 *
	70%	61.94 ± 32.35	0.69	26.84	−10.70 ± 24.62	0.07	0.38 *
1 min	0%	58.95 ± 30.35	0.63	29.03	−13.50 ± 25.70	0.01	0.45 *
	5%	58.9 ± 31.09	0.63	29.16	−13.59 ± 25.78	0.01	0.45 *
	10%	58.83 ± 30.93	0.63	29.21	−13.64 ± 25.81	0.01	0.45 *
	15%	58.69 ± 31.15	0.63	29.28	−13.70 ± 25.84	0.01	0.45 *
	30%	58.54 ± 30.53	0.63	29.46	−13.84 ± 26.00	0.02	0.45 *
	50%	58.04 ± 30.78	0.62	29.98	−14.18 ± 26.42	0.02	0.47 *
	70%	56.61 ± 31.31	0.60	30.98	−15.28 ± 26.94	0.04	0.52
30 s	0%	51.41 ± 28.62	0.53	34.18	−19.97 ± 27.74	−0.01	0.67
	5%	51.36 ± 28.7	0.53	34.21	−20.01 ± 27.88	−0.01	0.67
	10%	51.14 ± 28.54	0.53	34.36	−20.06 ± 27.91	−0.01	0.69
	15%	51.23 ± 28.86	0.52	34.48	−20.08 ± 27.93	−0.01	0.69
	30%	50.77 ± 28.80	0.52	34.73	−20.49 ± 28.04	−0.01	0.70
	50%	49.95 ± 29.05	0.51	35.42	−21.16 ± 28.41	0.00	0.71
	70%	48.11 ± 29.63	0.48	36.85	−22.63 ± 29.08	0.02	0.77

**Table 2 sensors-20-07122-t002:** Descriptive statistics (mean ± standard deviation) and statistical analysis of rMSSD for all of the time window and missing values. The statistical analysis is focused to compare 5 min HRV parameters to ultra-short ones. ** and * refer to the trivial and small ES of ultra-short HRV features compared to 5 min ones, respectively. r, RMSE, and ES refer to the correlation, the Root Mean Square Error and the effect size between the ‘gold standard’ and the ultra-short with and without missing values on rMSSD, respectively. Bias refers to mean ± standard deviation of the difference of ultra-short HRV parameters compared to ‘gold standard’ HRV parameters (i.e., 5 min time window), while systematic error is the Pearson’s correlation coefficient between means and differences between 5 min and ultra-short HRV estimation.

Time Window	Missing Values	rMSSD	r	RMSE	Bias	Systematic Error	ES
5 min	—	42.49 ± 20.75	—	—	—	—	—
4 min	0%	42.34 ± 22.12	0.93	7.87	−0.56 ± 7.85	0.16	0.03 **
	5%	42.18 ± 22.08	0.92	8.12	−0.58 ± 7.91	0.16	0.03 **
	10%	42.11 ± 22.06	0.92	8.48	−0.62 ± 8.18	0.18	0.03 **
	15%	42.07 ± 22.46	0.91	8.55	−0.67 ± 8.54	0.19	0.04 **
	30%	42.02 ± 22.96	0.90	9.97	−0.87 ± 9.93	0.21	0.05 **
	50%	41.35 ± 24.06	0.85	12.78	−1.50 ± 12.69	0.26	0.07 **
	70%	39.04 ± 26.09	0.72	18.34	−3.69 ± 17.96	0.32	0.16 **
3 min	0%	42.15 ± 22.36	0.92	8.82	−0.57 ± 8.81	0.19	0.04 **
	5%	42.05 ± 22.54	0.90	9.01	−0.66 ± 8.98	0.22	0.04 **
	10%	41.94 ± 23.01	0.89	9.56	−0.71 ± 9.06	0.23	0.04 **
	15%	41.78 ± 23.07	0.89	9.88	−0.85 ± 9.81	0.25	0.04 **
	30%	41.73 ± 23.21	0.88	10.97	−0.96 ± 10.93	0.24	0.05 **
	50%	41.08 ± 24.45	0.83	13.86	−1.56 ± 13.76	0.29	0.08 **
	70%	39.00 ± 26.89	0.71	19.21	−3.48 ± 18.89	0.36	0.16 **
2 min	0%	41.83 ± 22.70	0.90	9.85	−0.82 ± 9.82	0.19	0.04 **
	5%	41.79 ± 22.78	0.88	10.51	−1.16 ± 10.05	0.25	0.05 **
	10%	41.81 ± 22.18	0.88	10.36	−1.12 ± 9.98	0.25	0.05 **
	15%	41.79 ± 22.81	0.86	11.76	−1.16 ± 10.80	0.27	0.06 **
	30%	41.29 ± 23.60	0.86	12.05	−1.31 ± 11.98	0.25	0.07 **
	50%	40.40 ± 24.75	0.80	14.80	−2.17 ± 14.64	0.28	0.11 **
	70%	38.38 ± 27.60	0.68	20.63	−4.00 ± 20.24	0.38	0.17 **
1 min	0%	41.26 ± 23.47	0.86	12.02	−1.22 ± 11.96	0.24	0.06 **
	5%	40.97 ± 23.55	0.85	12.48	−1.50 ± 12.87	0.24	0.07 **
	10%	40.90 ± 23.61	0.84	12.69	−1.53 ± 13.13	0.26	0.07 **
	15%	40.69 ± 23.77	0.84	13.17	−1.71 ± 13.24	0.26	0.09 **
	30%	40.54 ± 24.57	0.81	14.42	−1.90 ± 14.29	0.29	0.09 **
	50%	39.52 ± 26.23	0.75	17.72	−2.84 ± 17.50	0.35	0.13 **
	70%	36.73 ± 29.70	0.62	24.01	−5.37 ± 23.40	0.44	0.22 *
30 s	0%	38.73 ± 24.13	0.79	15.13	−2.34 ± 14.95	0.30	0.11 **
	5%	38.71 ± 24.18	0.79	15.38	−2.55 ± 15.86	0.31	0.13 **
	10%	38.52 ± 24.67	0.78	15.99	−2.57 ± 15.97	0.31	0.13 **
	15%	37.74 ± 24.98	0.76	17.05	−3.13 ± 17.11	0.33	0.13 **
	30%	37.66 ± 25.34	0.73	17.83	−3.37 ± 17.51	0.33	0.15 **
	50%	36.07 ± 27.20	0.65	21.41	−4.83 ± 20.86	0.39	0.19 **
	70%	32.63 ± 30.47	0.53	27.14	−7.80 ± 26.00	0.47	0.31 *

**Table 3 sensors-20-07122-t003:** Spearman’s rank correlation coefficient between HRV features in 5 min time series.

	SDNN	rMSSD
VLF	0.62	0.19
LF	0.51	0.43
HF	0.45	0.80
LF/HF	−0.04	−0.47
Total Power	0.73	0.45

**Table 4 sensors-20-07122-t004:** Descriptive statistics (mean ± standard error of the mean) of VLF, LF, HF, LF/HF, and Total power of the power spectrum density for all of the time window and missing values. The statistical analysis is focused to compare 5 min HRV parameters to ultra-short ones. ** and * refer to the trivial and small ES of ultra-short HRV features compared to 5 min ones, respectively.

Time Window	Missing Values	VLF	LF	HF	LF/HF	Total Power
5 min	—	1891.44 ± 46.10	1208.11 ± 30.84	689.49 ± 24.84	3.31 ± 0.04	3907.50 ± 86.95
4 min	0%	1533.88 ± 37.96 **	1208.11 ± 30.84 **	689.48 ± 24.84 **	3.31 ± 0.04 **	3549.94 ± 80.76 **
	5%	1462.12 ± 35.36 **	1183.27 ± 31.12 **	736.11 ± 25.53 **	2.72 ± 0.03	3496.31 ± 80.57 **
	10%	1397.1 ± 34.58 **	1147.97 ± 30.16 **	782.07 ± 25.4 **	2.31 ± 0.02	3438.69 ± 79.8 **
	15%	1326.12 ± 33.39 **	1112.15 ± 24.97 **	828 ± 24.78 **	2.02 ± 0.02	3374.25 ± 71.88 **
	30%	1128.62 ± 27.88 *	1027.24 ± 17.10 **	954.92 ± 22.97 *	1.44 ± 0.01	3208.93 ± 59.54 **
	50%	853.10 ± 22.20 *	917.61 ± 19.01 **	1123.03 ± 30.88 *	0.99 ± 0.01	2976.95 ± 66.27 **
	70%	567.70 ± 15.31	819.50 ± 27.86 **	1313.26 ± 32.97 *	0.69 ± 0.004	2778.61 ± 73.01 **
3 min	0%	1127.11 ± 25.10 *	1110.41 ± 19.26 **	547.88 ± 25.18 **	3.41 ± 0.04	2897.76 ± 58.20 **
	5%	1079.4 ± 24.31 *	1083.32 ± 18.99 **	591.89 ± 26.46 **	2.81 ± 0.03	2861.37 ± 59.38 *
	10%	1023.22 ± 22.57 *	1047.71 ± 15.22 **	625.92 ± 17.50 **	2.4 ± 0.02	2799.93 ± 46.28 *
	15%	975.01 ± 21.49 *	1020.62 ± 16.45 **	666.64 ± 18.70 **	2.09 ± 0.02	2761.14 ± 48.79 *
	30%	827.45 ± 18.66 *	937.27 ± 13.08 **	785.34 ± 13.43 **	1.48 ± 0.01	2640.77 ± 40.29 *
	50%	625.49 ± 13.81	822.96 ± 12.18 *	960.21 ± 17.47 *	1.01 ± 0.01	2484.46 ± 40.24 *
	70%	429.55 ± 9.17	719.21 ± 13.36 *	1137.89 ± 20.49 *	0.70 ± 0.004	2350.27 ± 41.60 *
2 min	0%	1035.59 ± 24.53 *	1082.92 ± 18.27 **	518.80 ± 16.84 **	3.42 ± 0.04	2743.70 ± 48.45 *
	5%	989.72 ± 23.62 *	1054.75 ± 17.79 **	554.49 ± 18.46 **	2.86 ± 0.03	2702.02 ± 49.17 *
	10%	946.56 ± 22.56 *	1028.3 ± 17.47 **	592.71 ± 16.89 **	2.47 ± 0.03	2667.16 ± 47.16 *
	15%	893.91 ± 21.12 *	997.58 ± 16.83 **	639.02 ± 19.08 **	2.14 ± 0.02	2626.85 ± 48.55 *
	30%	753.04 ± 17.71	904.07 ± 14.64 **	743.62 ± 14.54 **	1.51 ± 0.01	2486.35 ± 41.52 *
	50%	572.83 ± 13.52	786.10 ± 12.41 *	895.60 ± 15.57 *	1.02 ± 0.01	2326.40 ± 38.32 *
	70%	394.79 ± 9.66	682.95 ± 14.42 *	1067.63 ± 28.46 *	0.71 ± 0.005	2204.56 ± 50.53 *
1 min	0%	616.19 ± 16.25	1060.77 ± 17.50 **	491.83 ± 10.75 **	3.46 ± 0.04	2272.02 ± 38.02 *
	5%	588.58 ± 15.48	1033.71 ± 16.93 **	527.44 ± 11.94 **	2.93 ± 0.03	2249.53 ± 38.05 *
	10%	559.61 ± 14.65	999.09 ± 16.35 **	557.29 ± 9.69 **	2.51 ± 0.03	2212.12 ± 35.94 *
	15%	534.03 ± 13.78	964.84 ± 15.65 **	589.81 ± 10.8 **	2.21 ± 0.02	2181.41 ± 35.56 *
	30%	453.45 ± 11.82	880.46 ± 14.18 *	693.12 ± 12.15 **	1.57 ± 0.02	2110.12 ± 34.48 *
	50%	337.15 ± 8.56	741.32 ± 12.09 *	831.16 ± 14.37 **	1.05 ± 0.01	1976.68 ± 32.74 *
	70%	226.16 ± 5.78	623.91 ± 11.57 *	964.02 ± 17.28 *	0.71 ± 0.005	1867.88 ± 33.62
30 s	0%	—	1043.22 ± 19.46 **	497.03 ± 10.04 **	3.47 ± 0.05	1634.33 ± 27.33
	5%	—	1009.19 ± 19.01 **	518.67 ± 10.16 **	3.01 ± 0.04	1618.65 ± 27.22
	10%	—	973.86 ± 18.21 **	536.36 ± 9.59 **	2.64 ± 0.04	1597.54 ± 26.45
	15%	—	932.54 ± 17.52 **	571.69 ± 10.78 **	2.29 ± 0.03	1588.48 ± 26.87
	30%	—	820.17 ± 15.69 *	628.70 ± 11.05 **	1.65 ± 0.02	1521.87 ± 26.04
	50%	—	671.04 ± 13.42 *	720.85 ± 13.05 **	1.08 ± 0.01	1451.23 ± 25.83
	70%	—	539.98 ± 11.27	839.28 ± 16.60 *	0.72 ± 0.01	1425.51 ± 27.72

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
