# Peer review of "Error Estimation of Ultra-Short Heart Rate Variability Parameters: Effect of Missing Data Caused by Motion Artifacts"

_sensors, 2020, doi:10.3390/s20247122_

Round 1
Reviewer 1 Report
Dear Authors,
I have an opportunity to review a manuscript entitled “Error estimation of ultra-short heart rate variability parameters: effect of missing data caused by motion artefacts”.
I find this manuscript interesting and good written however I have some doubts and suggestion that I would like to have clarified or answered.
Major:
- Generally, it is interesting how big the error of ultra-short HRV is, but I think that practical utility of its analysis with wrist-worn low-cost devices is limited. Technology in ECG acquisition (even with small devices) is very advanced. I recommend rearrange towards modest paragraphs which suggest that this technology can be used instead of good quality simple ECG (for example one lead). I personally think that it’s better to improve noise reduction than extrapolate missing data. Nevertheless I think that conceptually the main idea of the paper is interesting.
- It would be interesting add as reference 24-h HRV analysis. If it is possible please do it. If not, consider Study limitation Section or paragraph and put an information that it is not performed.
- I think that equation (1) and (2) is widely known and therefore not needed to be presented in the manuscript. Consider removing it and leave only appropriate reference.
- There is Bland and Altman plot decribed but I cannot find any in manuscript either in supplemental material. Is there any particular reason mentioning BA plots in the text?
- Table 1. There is mentioned that single and double asterix mean(…), but there is no single value with double asterix. If so remove this explanation of two asterix or maybe some values should have double asterix. Please explain.
- In spectral analysis one of the most important value is HF/LF ratio. It shoud be given in Table 2. Please provide it. If it is impossible please explain why, and add it to the limitation of the study section/paragraph.
- The study is performed on young males – it should be added to limitation of the study.
Minor:
- IBI – abbreviation was explained in the Method section, but it had been used earlier in Introduction. Ought to be explained on the first use.
- In Table 1. There is no precise explanation of rmse and bias. Should be provided.
- Statement in lines 208 – 211 needs reference.
I am looking forward to reading a corrected manuscript.
Best regards,
Reviewer 2 Report
- In lines 16 to 18 the text is difficult to understand: "Finally, the missing values ​​low affect both rMSSD and SDNN slowing increase the estimation error as the percentage of missing values ​​increase". Needs to be rewritten.
- In lines 30 to 32, it is quoted that "In particular, high HRV values ​​have been found to be strongly correlated to increased risk of mortality (particularly among the elderly) causing for example atrial fibrillation". In fact, what is widely known by the literature is that low HRV values ​​are associated with the occurrence of severe arrhythmias and death. Reference 7 cited (Stein et al. J Cardiovasc Electrophysiol, Vol. 16, pp. 954-959, September 2005) does not explicitly state that high HRV values ​​are associated with a higher risk of mortality. Those authors hypothesized that "...a higher prevalence of abnormal hourly heart rate patterns would be associated with increased HRV. We further hypothesized that a higher prevalence of abnormal heart rate patterns might be associated with mortality after adjustment for age and gender, factors known to be associated with mortality in the elderly ". In this way, they found that there was a different behavior in the hourly pattern with greater dispersion of values ​​in cases with poor evolution, but not that the average values of individual variables ​​were sistematically above normal values. Please double check that statement.
- The authors here considered the missing ranges as 0, 30%, 50% and 70%. In the real world, series with more than 5% of artifacts are generally disregarded as being inadequate for HRV assessment. The presente authors did not study such low ranges, but even at 30% the effects were not so negative. Therefore, why use such high values of missing instead of lower values, such as 0, 5%, 10% and 15%? Daily experience shows us that 10% of missing, already has a great impact on the visual aspect of the tachogram. I believe that no researcher would accept to include in their analysis cases with artifact rates of 30% or more. Please comment about this.
- Based on the data presented by the authors, the SDNN evaluation suggests that 2 minutes of time series length seems to be the minimum acceptable time for use, because with less than that the correlations fall below 0.7 even with 0% missing and the effect size exceeds the 0.3 mark. For the RMSSD, the limits would be more flexible, but as a suggestion for standardization for a suitable minimum duration for studying the set of commonly used variables, could the time of 2 minutes be recommended? This could be a recommendation in the study Discussion.
Round 2
Reviewer 1 Report
Dear Authors,
- All my doubts and suggestions were adequately addressed.
- I have no other issues regarding to this manuscript.
- I recommend to accept the manuscript for publication.
Best regards.
Author Response
We would like to thanks the reviewer for his useful comments that have permitted to improve our work.
Reviewer 2 Report
The changes included in the text adequately followed what was suggested. I do not need further clarification
Author Response

(The authors gave the same response as above.)
